# A Cross-Sectional Reproducibility Study of a Standard Camera Sensor Using Artificial Intelligence to Assess Food Items: The FoodIntech Project

**DOI:** 10.3390/nu14010221

**Published:** 2022-01-05

**Authors:** Virginie Van Wymelbeke-Delannoy, Charles Juhel, Hugo Bole, Amadou-Khalilou Sow, Charline Guyot, Farah Belbaghdadi, Olivier Brousse, Michel Paindavoine

**Affiliations:** 1Elderly Unit, University Hospital Center Dijon Bourgogne F Mitterrand, F-21000 Dijon, France; charline.guyot@chu-dijon.fr (C.G.); farah.belbaghdadi@chu-dijon.fr (F.B.); 2Centre des Sciences du Goût et de l’Alimentation, INRAE, Université de Bourgogne Franche-Comté, CNRS, Agrosup, F-21000 Dijon, France; 3ATOL Conseils & Développements (AtolCD), ZAE les Terres d’Or, Route de Saint Philibert, F-21220 Gevrey-Chambertin, France; cju@atolcd.com (C.J.); hbo@atolcd.com (H.B.); 4CHU Dijon Bourgogne, Inserm, Université de Bourgogne, CIC 1432, Module Épidémiologie Clinique, F-21000 Dijon, France; amadou-khalilou.sow@chu-dijon.fr; 5Yumain, 14 Rue Pierre de Coubertin, F-21000 Dijon, France; olivier.brousse@yumain.fr (O.B.); michel.paindavoine@yumain.fr (M.P.)

**Keywords:** artificial intelligence, portion evaluation, mobile phone images, machine learning, reliability

## Abstract

Having a system to measure food consumption is important to establish whether individual nutritional needs are being met in order to act quickly and to minimize the risk of undernutrition. Here, we tested a smartphone-based food consumption assessment system named FoodIntech. FoodIntech, which is based on AI using deep neural networks (DNN), automatically recognizes food items and dishes and calculates food leftovers using an image-based approach, i.e., it does not require human intervention to assess food consumption. This method uses one-input and one-output images by means of the detection and synchronization of a QRcode located on the meal tray. The DNN are then used to process the images and implement food detection, segmentation and recognition. Overall, 22,544 situations analyzed from 149 dishes were used to test the reliability of this method. The reliability of the AI results, based on the central intra-class correlation coefficient values, appeared to be excellent for 39% of the dishes (*n* = 58 dishes) and good for 19% (*n* = 28). The implementation of this method is an effective way to improve the recognition of dishes and it is possible, with a sufficient number of photos, to extend the capabilities of the tool to new dishes and foods.

## 1. Introduction

Measuring patient food consumption and food waste are important for the goals of healthy eating and sustainability. In hospitals, having a system to measure food consumption is a key element to know if patients’ nutritional needs are being properly covered and to decrease food waste. However, recording food intake is challenging to implement in the hospital context, and often suboptimal [1,2], and nutritional monitoring is thus rarely part of the clinical routine [3]. Potential barriers include the lack of knowledge and poor awareness of caregivers, the medical prescription of the three-day food record, and the absence of a quick and easy-to-use monitoring tool that is accurate and precise. The most used method is a direct visual estimation [4]. This method consists of quantifying the remaining part of the food intake of subjects during mealtimes by trained staff (care staff, dieticians or experimenters). Direct visual estimation is easy to organize, and has proven to be reliable in different institutions, including geriatric units [5,6,7]. This method can be used to differentiate each item of food served and leftovers by portion (1, ¾, ½, ¼ or 0 of food served) because employees can mentally separate different items on a plate. However, direct visual estimation is often performed quickly and with a low degree of precision because staff are not always trained. In addition, the estimation of the food before and after consumption is not always done by the same employee. This approach is also subjective, which may lead to errors. Additionally, while a large number of subjects can be followed, this method also requires the availability of multiple employees at the same time. 

In a clinical research context, the weighing method remains the “gold standard” to measure actual food consumption. This method is based on the difference between the weights of foods offered before consumption and those not consumed after consumption by the participant [8,9]. However, the weighing method is complex in real-life in facilities, and the delivery modes are difficult to reconcile with data requirements. This method is time-consuming, requires significant staff resources, and remains unsuitable for the hospital environment [8,9]. 

Other methods using photography have been developed. Martin et al. developed the Remote Food Photography Method (RFPM), which involves participants capturing images of their food selection and leftovers [10,11]. These images were then sent to the research center via a wireless network, where they are analyzed by dietitians to estimate food intake. These same authors were able to show that there was a good correspondence between the RFPM method and the weighing method, with a difference ranging from −9.7% for the starter to 6.2% for the vegetables [12]. This technique does not require a large number of personnel on site during mealtimes. It was compared to the visual estimation process in collective dining facilities [13], to the weighing method in an experimental restoration situation [14], in a cafeteria situation [15] and in a geriatric institution [16]. It avoids the bias associated with the presence of experimenters at mealtimes and, by capturing images, the analyses can be repeated by different people. Although these methods are reliable, they all require a professional to perform the data analysis, and they remain time-consuming.

Several novel techniques have recently been proposed thanks to advances in Artificial Intelligence (AI) applications. AI has expanded in different domains using images with new opportunities in nutrient science research [17]. In a review, mobile applications based on systems using AI were of significant importance in the different fields of studies on biomedical and clinical nutrients research and nutritional epidemiology. Among the available AI applications, two algorithms can be used: machine learning (ML) algorithms, widely used in studies on the influence of nutrients on the functioning of the human body in health and disease; and deep learning (DL) algorithms, used in clinical studies on nutrient intake [17]. ML is an AI domain related to algorithms that improve automatically through gathered experience, making it possible to create mathematical models for decision-making. DL is a subtype of ML, with the advantage of program autonomy that can build functions used in recognition. Recently, Lu et al. proposed a dietary assessment system, named goFOOD^TM^, based on AI using DL [18]. This method uses the deep neural networks to process two images taken by a single press of the camera shutter button. Even if the results demonstrated that goFOOD^TM^ performed better with detection using DL algorithms than experienced dietitians did, this method still requires human intervention.

In this paper, we propose a smartphone-based food consumption assessment system, called FoodIntech. FoodIntech, which is based on AI using DL, automatically recognizes food items and dishes and calculates portion leftovers using an image-based approach, i.e., it does not require human intervention to assess food consumption. This method uses one-input and one-output images by means of the detection and synchronization of a QRcode label stuck to the meal tray. Then, the deep neural networks are used to process the images and implement food detection, segmentation and recognition. The aim of this paper is to test the reliability of this method in laboratory conditions, but as similar as possible to routine clinical practice in a hospital environment.

## 2. Materials and Methods

### 2.1. General Procedure

The tool developed is the subject of an industrial program whose technical details cannot be fully disclosed.

The implemented method is based on the analysis of food tray pictures taken by a standard out-of-the box mobile phone (i.e., Android Samsung S8), before and after the simulated consumption of a standard meal tray taken from the central kitchen of the Dijon University Hospital, and processed by laboratory research staff. However, the method implemented here attempted to emulate routine clinical practice in a hospital environment: the smartphone was attached by a system adapted to the heating trolley used to distribute the meal trays in the hospital, the food quantities measured were identical to those served to patients, and the same dishes, the same trays, the same crockery and the same arrangement of dishes were also used. The protocol was designed to simulate patient consumption in a large range of situations to build a consolidated vision of the liability and repeatability of the process (detailed in Weighed food method paragraph). 

Each pair of pictures of the tray (before/after) was recorded in an experimental database and linked to the weight values of each food item on each plate corresponding to the captured images. The food items were weighed before and after the simulation by the research team, for all the plates and in each of the experimental conditions established in the protocol. The collected data constitute the reference dataset for comparison of the study on which the technical process of the solution has been evaluated. 

AI based on deep learning received the input of 13,152 pictures showing 26,584 food item consumption situations produced throughout the study to ensure learning based on image segmentation. The AI program therefore needed a very large number of pictures per food item or dish to learn to recognize it, around 200 images for each.

The raw data returned by AI cannot immediately be processed by the system; it has to be interpreted. As such, a transcoding overlay was added. The sole purpose of the overlay is to process the AI results and map the using the list of known dishes from the menu.

Sometimes, the transcoding overlay is unable to map the dishes properly, especially when the AI returns a relatively vague result (e.g., the same element several times). In this case, a refinement increment is required to be able to discern which is which, and to return an accurate response, enabling a flawless transcoding process.

The results of the system were also challenged to obtain a precise percentage of each food component portion remaining from one picture to another compared to the weighted method.

Each new increment of the AI was thoroughly tested to track any possible regression. To do so, each dataset line result was reprocessed and then compared with the experimental value to produce a comprehensive vision of the AI’s capabilities for this version.

Both the AI core and the transcoding overlay were modified in the process so they could deliver the best results. Many iterations were made, focusing on different priorities each time, to be able to come up with a satisfying AI result. Monitoring the different results release after release made it possible to follow the evolution of the AI and to establish the optimal process of teaching the neural network to recognize new dishes/components. This also made is possible to manage inconsistencies with the transcoding system to achieve a technically reliable system capable of digital detection and weighing (not in this paper).

### 2.2. Algorithm Used for Deep Learning 

The measurement of the quantity of food ingested was based on an analysis of images taken before and after meals. This analysis consisted of a measurement of the pixel surface of the different food items present in the menu. Thanks to the progress that has been made over the last few years with deep learning techniques, image analysis has also become increasingly powerful. Three main approaches are used to analyze images with these techniques: the first one, called “Classification”, indicated the presence or not of a particular object (here food) in the analyzed image. The second more precise approach, called “Detection”, indicated in which area of the image the recognized object was located. Finally, the third approach, which is the most precise, consisted in a fine clipping of the detected object. This approach, called “Instance Segmentation”, was well adapted to our application. 

There are mainly two types of Deep Learning neural networks dedicated to instance segmentation: U-Net [19] and Mask R-CNN [20]. Based on study of Vuola and al, we decided to use Mask R-CNN rather than U-Net for its better ability to identify and separate small and medium-sized objects that compose a large part of the food items present in the analyzed plates [21].

For both training and inference, the spatial resolution of the images was between 800 and 1365 pixels per side. The Mask-RCNN neural networks were trained and used on a computer equipped with a NVidia Titan RTX graphics card with 24 GB VRAM.

For the implementation of Mask-RCNN, we used the mask_rcnn_inception_resnet_v2_atrous_coco one from Tensor Flow Object Detection Model Zoo [22]. It is based on the inception-resnet-v2 backbone [23], and provides great precision masks at the cost of a high computing power requirement. Classical data augmentation was used for the training with a randomized image horizontal flip and a randomized brightness adjustment at each period of the training. In order to achieve satisfactory performance, it was necessary to train the Mask-RCNN with an average of 200 examples per food item; items were accurately annotated and included in the training database. 

The details of the learning parameters are provided in Appendix B.

Figure 1a,b gives an illustration of this segmentation technique. The left part of the figure corresponds to the image of the plate acquired with the mobile phone and the right part to the image of the plate analyzed with the Mask-RCNN segmentation algorithm. We can see that the different foods contained in the plate are very well separated. From this segmentation, the surface of each food item is measured by the number of pixels needed to cover the item.

### 2.3. Data Collection and Process or Procedure of Deep Learning View Synthesis Approach 

The Foodintech procedure of using deep learning synthesis to determine food portions is shown in Figure 2:

(1) A standard mobile phone was used, and the camera zoom of the device allowed to automatically focus the frame around the border of the tray; (2) and (3) two captured images before and after simulated consumption were taken and synchronized thanks to the QRcode label on the tray; (4) the bounding boxes show the results of Mask R-CNN to identify food items; (5) detection of food items: precision enhanced by deep learning annotations; (6) identification of food items: each food item is identified among 77 food categories, the exact recipe is defined by the menu which is provided in advance; (7) solving conflicts: transcoding layer applies decision rules to AI results conflicts or errors between items; (8) segmentation counting the numbers of pixels corresponding to different regions on the plate; (9) digital weighing: food intake calculated by % of missing portion for each item; (10) results dataset: displays consumption % of each plate on each tray applied to known served portion weight to show results in grams of food intake compared to real portion weight measures.

Dish preparation*—*The dishes were prepared in the central kitchen of the Dijon University Hospital according to the standardized recipes and the same procedures for each day of measurement. The plates were prepared in a meal tray as if they were being served to patients, with the same quantities, presentation and daily menus. The quantities served adhered to the nutritional guidelines for adults in France. The captured images of 169 different labels of dishes were used to test the reliability of the FoodIntech method. They were produced between 22 December 2020, and 15 May 2021, at the Dijon University Hospital.

Weighed food method*—*Each meal component was weighed before and after each simulated consumption using the same electronic scales (TEFAL, precision +/− 1 g). The experimenter, always the same person, initially weighed the dish with the quantities of the nutritional guidelines provided by the central kitchen and then removed 1 spoonful at a time until there was nothing left on the plate. The obtained weight allowed us to define the different experimental conditions for each dish. Then, the experimental conditions were reproduced between 4 to 14 different simulated consumptions. Each food component was weighed between 5 to 20 times for each experimental condition, and 30 to 200 pictures were taken per food item depending on the initial quantity served.

Food Image acquisition*—*Each meal tray was pictured before and after experimental conditions were applied using a standard mobile smartphone (Samsung Galaxy S8) with a zoom function on the camera. To estimate the reliability of the AI algorithm results, i.e., the estimated consumption percentage of a given dish, each of the 169 dishes studied had to be photographed 4 to a maximum of 14 times. A total of 13,152 pairs of images of 26,584 food items in various consumption conditions were produced. Two captured images, one input and one output, were synchronized thanks to a QRcode label (Figure 3a,b). 

### 2.4. Statistical Analysis

The sample size of this study was determined using the PASS software. A sample size of 20 subjects (camera angles) with 14 repetitions (food conditions) per subject achieves 80% power. This can detect an intra-class correlation of 0.90 under the alternative hypothesis when the intraclass correlation under the null hypothesis is 0.80, using an F-test with a significance level of 0.05 [24].

The estimation of the reliability of the AI results for each dish was assessed by the Type 3 Intra-class Correlation Coefficient (ICC) [25]. We used type 3 ICC issued from a two-way mixed model as recommended by Koo and Li 2016 [26]. Indeed, the reliability of a measurement refers to its reproducibility when it is repeated on the same subject. We wanted to have a complete overview with all possible camera angles, so positions where randomly selected among all the possible camera angles (infinite population). This corresponded to the random subject effect. Concerning repetitions, we were interested in assessing the 14 possible conditions, which reflect the finite, well-defined, possible measurement circumstances, and they were not randomly selected in an infinite population. This corresponded to the repetition factor, which was considered as a fixed effect in our model. All of these considerations were applied for each given food.

These ICCs were determined using a random effect model from the irrNA (R package version 0.2.2; Berlin Germany) R software package (version 4.1.0, R Core Team; R Foundation for Statistical Computing, Vienna, Austria). It is important to note that the calculation of the ICC does not require the normality of the measurements and that the package allows the estimates to be made despite a varying number of conditions and images. The values of the ICC are between 0 and 1. The closer the ICC is to 1, the more similar the results are using AI under the same condition; however, the lower the ICC is (close to zero), the more the results of the AI diverge for the same condition. The central estimates for each ICC are accompanied by their 95% confidence interval, providing an estimate of the range in which the true value of the ICC could lie at the risk of a type I error of 5%. According to Shrout and Fleiss, an ICC greater than or equal to 0.8 supports excellent reliability; an ICC between 0.7 and 0.8 indicates good reliability. Values below 0.7 indicate moderate reliability at best [25,27].

## 3. Results

Dishes for which data was only available for one condition, or dishes for which measurements were not available for all conditions were excluded (exclusion of 20 dishes and 15% of all the pictures/situations, i.e., 4040 pictures). Thus, 22,544 analyzed situations for 149 dishes were considered for this reproducibility analysis. In total, based on the central ICC values obtained and the classification proposed by Shrout and Fleiss, the reliability of the AI results appeared to be excellent for 39% of the dishes (*n* = 58 dishes) and good for 19% (*n* = 28). The reliability appeared insufficient for 42% of the dishes (Figure 4). The ICCs for each dish are available in Appendix A (Table A1).

An additional analysis was performed for dishes for which the number of pictures was greater than or equal to 200 (*n* = 60 dishes). The ICCs obtained were in favor of excellent reliability for 45% of the dishes selected (27/60), of good reliability for 17% of the dishes (*n* = 10) and for insufficient reliability for 38% of the dishes.

## 4. Discussion 

In this paper, we report our testing of FoodIntech, which is a dietary portion assessment system that estimates the consumed portion size of a meal using images automatically captured by a standard smartphone. This study is a pioneer study including 149 different labels of dishes, with single or composite food items. We covered high volume images of 22,544 food items using automatic detection and AI. The images were taken in experimental conditions approaching routine clinical practice in a hospital environment: a smartphone was attached to a heating trolley used to distribute meal trays, the quantities of food were identical to those served to patients, and we used the same dishes, trays, crockery, and arrangement of dishes on the trays.

A review of the literature shows that there is has been real interest in using image-based dietary assessments in different situations: in free-living individuals [10,28], and in specific environments such as hospitals, [29,30,31], laboratories, [10,32] and cafeterias [33]. This method is often combined with food records or voice recording describing served and consumed meals [28,34], or associated with video recordings to be compared to weighed food records in free-living young adults [35]. However, even if the performance was found to have good reliability compared to traditional methods by some researchers, these image-based dietary assessments require the capture of images by participants and/or trained staff members to observe or calculate the individual’s consumption [10,29,31,36,37].

The main advantage of AI is that human intervention is not required, and the results are obtained instantly. This is why the automatic detection of food in real-life contexts with data acquired by a wearable camera or smartphone in association with an AI analysis is challenging: image quality depends on variable factors such as appropriate lighting, image resolution, and limited blur caused by the movement of person taking the pictures. Many studies have demonstrated the validity of AI to automatically detect food items. To our knowledge, none have tested reproducibility on a number of complex foods or dishes as large as in our study. In addition, few studies have specified the number of analyzed images or the conditions under which the images were analyzed. Ji et al. [38] assessed the relative validity of an image-based dietary assessment app—Keenoa—and a three-day food diary in a sample of 102 healthy Canadian adults, but no information was given about the sample of images used and the authors showed that the validity of Keenoa was better at the group level than the individual level. In Fang et al. [39], the authors estimated food energy based on images and the generative adversarial network architecture. They validated the proposed method using data collected by 45 men and women between 21–65 years old, and obtained accurate food energy estimation with an average error of 209 kilocalories for eating occasion images (1875 paired images) collected from the Food in Focus study using the mobile food record. However, the authors specify the need to combine automatically detected food labels, segmentation masks, and contextual dietary information to further improve the accuracy of their food portion estimation system. Jia et al. [40] developed an artificial intelligence-based algorithm which can automatically detect food items from 1543 food-images acquired by an egocentric wearable camera, called the eButton. Even in the absence of reproducibility data, they reported that accuracy, sensitivity, specificity and precision were 98.7, 98.2, 99.2 and 99.2%, respectively for food images. In a recent publication, Lu et al. [18], proposed goFOODTM as a dietary assessment system based on AI to estimate the calorie and macronutrient content of a meal with food images captured by a smartphone. goFOODTM uses deep neural networks to process the two images and implements food detection, segmentation and recognition, while a 3D reconstruction algorithm estimates the food’s volume. Thus, the calorie and macronutrient content was calculated from 319 fine-grain food categories, and the authors specified that the validation was performed using two multimedia databases containing non-standardized and fast food meals (one contained 80 central-European style meals and another one contained 20 fast-food type meals). However, in this study, there was no indication of reproducibility or the number of images analyzed. Other authors showed the error estimation of their system: Lu et al. [41] showed an estimation error of 15% with their system allowing a sequential semantic food segmentation and estimation of the volume of the consumed food with 322 meals. Sudo et al. [42] showed an estimation error close to 16.4% with their system presenting a novel algorithm that can estimate healthiness from meal images without requiring manual inputs. 

It is still impossible for the system to recognize all the food categories in the world and in real-life, and this remains a limit for all existing applications. The limitation of our system is mainly determined by the training data.

We showed that 86 dishes were correctly detected by the system in repeatable conditions over an ICC value of 0.7/1. The 63 non-repeatable dishes (ICC < 0.7) show the limitations of the system: 10 were served in containers (cups or tubes), 18 had more than 200 pictures but were specially shaped food items or dishes (e.g., oranges or other fruits with skin leftover) and 35 were reproduced in less than 200 pictures. In order to address the limitations associated with food recognition, we will try to analyze their origins and the possible solutions:

-First of all, is the AI able to improve its results on the identified less detected food items and dishes? If we look precisely at the identified items that were the least reliable in the field of the study (for example yogurts or purées in containers) it appears that it was difficult to build a volumetric vision from a single sight view at 90°. It produces, with containers that are higher than wide, a drop shadow that leads to the difficult identification of height and volume. This issue could probably be improved by the use of new virtual 3D sensors embedded in recent market smartphones based on, for example, the Time of Flight (ToF) technology proposed by Samsung from the Galaxy S10 version. This technology calculates the speed of photons to access the surface of the target objects in order to construct a virtual depth vision of the objects in three dimensions. Additionally, some of the products might be transferred to glass or plastic containers in order to avoid the identified issues. Lo et al. [43] created an objective dietary assessment system based on a distinct neural network. They used a depth image, the whole 3D point cloud map and iterative closest point algorithms to improve dietary behavior management. They demonstrated that the proposed network architecture and point cloud completion algorithms can implicitly learn the 3D structures of various shapes and restore the occluded part of food items to allow better volume estimation. -Secondly, is the AI able to improve its results on the fruit with a peel? Inedible leftovers are currently interpreted as the fruit itself in most cases. However, we are confident that the AI can learn to recognize these leftovers from the flesh of the fruit with new learning. In addition, we can code some specific rules to the transcoding overlay to help with the reproducibility of correct identifications.-Finally, is the size of the sample enough, especially when we consider that some food items with more than 200 pictures in records had insufficient reliability? Some dishes, as well as some food containers, are more difficult to recognize and segment than others due to their shape, colors or ingredients, with the mixture on the plate making them more difficult to identify. Like human vision, computer vision has limitations that will never achieve 100% performance. However, just like humans, AI can improve the recognition of certain dishes or situations through a wider learning process and thus increase the number of reference pictures and segmentations. The method validated in this study, in particular, can obtain high-performance results for complex food items, allowing us to extrapolate that significant improvements could be obtained for dishes that are still poorly recognized or poorly reliable. Examples of these complex items include the “Colombo of veal with mangoes”, which had an ICC of 0.897 with 201 photos, the “Nicoise salad”, which had an ICC of 0.899 with 199 photos, and the “gourmet mixed salad”, which had an ICC of 0.949 with 198 photos.

## 5. Conclusions

These results were obtained using different learning steps, demonstrating that the method used to improve the recognition of dishes is effective, and that with a sufficient number of photos it can be extended to new dishes and foods. This study is, to our knowledge, one of the first to have tested such a large sample (22,544 images), and we obtained a high level of precision for more than half (57.8%) of a wide range of foods (149). To our knowledge, this method represents a paradigm shift in dietary assessment. AI technology can automatically detect foods with camera-acquired images, reducing both the burden of data processing and paper transcription errors. An additional benefit of AI is the ability to immediately analyze the data and obtain results. This study demonstrates that this image recognition technique could be an exploitable clinical tool for monitoring the food intake of hospital patients.

## 6. Future Work

Future work will aim to show the validity and the usability of the FoodIntech artificial intelligence system for the evaluation of food consumption in hospitalized patients. The system will provide a measure of patient food consumption, which could then be associated with the patient’s age, gender and body weight. Dietitians or physicians could use this information to adapt in-hospital menus to patient needs.

## Figures and Tables

**Figure 1 nutrients-14-00221-f001:**
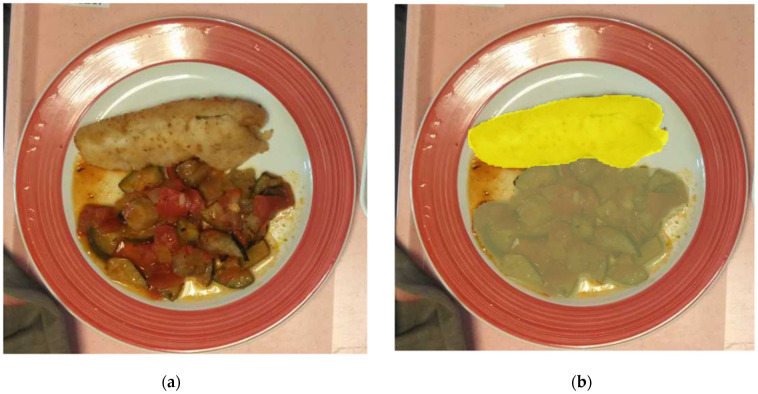
Food segmentation example applied to (**a**) plate with food served, (**b**) segmentation of each food of the plate.

**Figure 2 nutrients-14-00221-f002:**
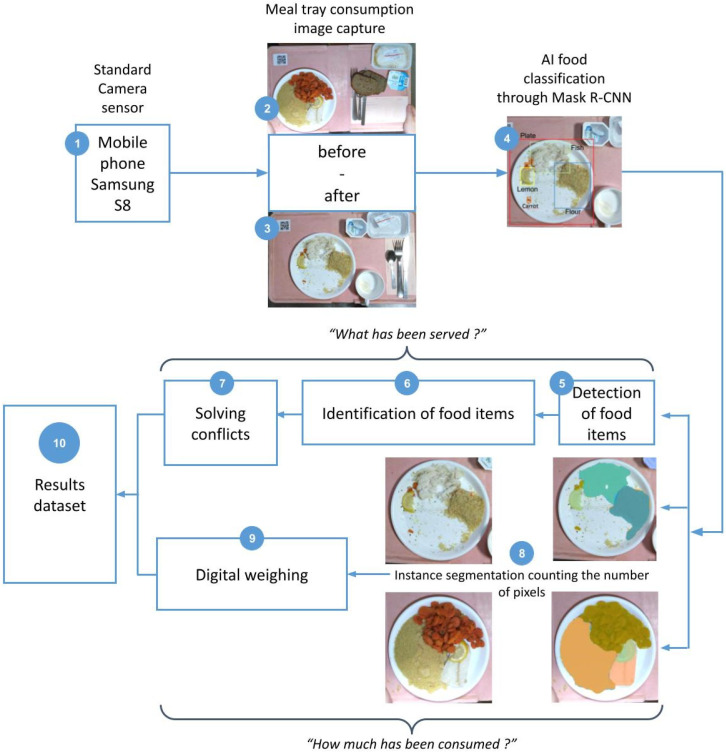
Implementing computer vision technology and a deep learning process to recognize food items and calculate the amount of missing food between two meal tray pictures.

**Figure 3 nutrients-14-00221-f003:**
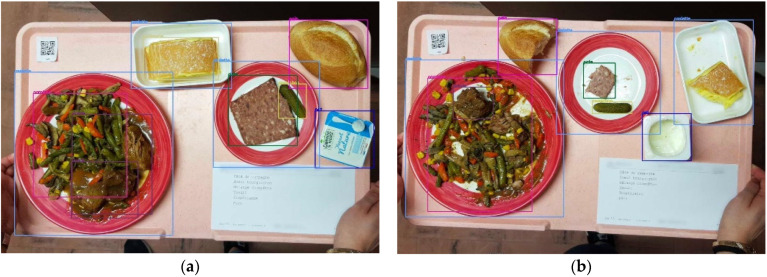
Meal trays with QRCode label (**a**) before consumption and (**b**) after consumption.

**Figure 4 nutrients-14-00221-f004:**
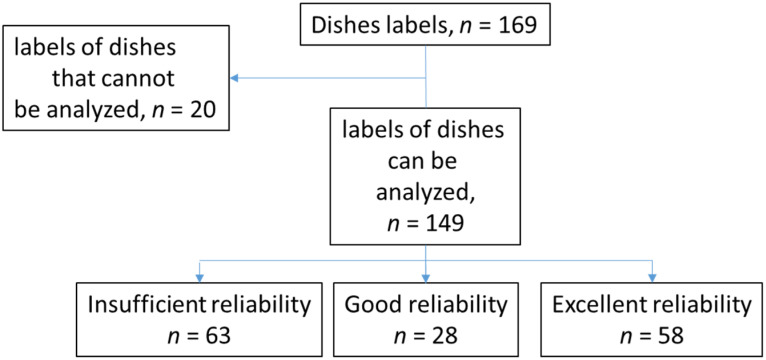
Flow chart of the design.

## Data Availability

Not applicable.

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
