# Peer review of "A Cross-Sectional Reproducibility Study of a Standard Camera Sensor Using Artificial Intelligence to Assess Food Items: The FoodIntech Project"

_nutrients, 2022, doi:10.3390/nu14010221_

Round 1

Reviewer 1 Report

This is a pioneer study including 169 different labels of dishes were covered high volume images. Especially imply AI technology to assess patient food consumption. Various labeled characteristics were recorded and compared among food intake. Type 3 Intra-class Correlation Coefficient (ICC) results were made regarding different labeled characteristic. They found that reliability of the AI appeared to be excellent for the recognition of dishes.

This is an interesting study with some new findings in this area of research. The sample size of subjects is small for analysis. However, I nevertheless have the following comments that required to be addressed.

  1. The study design should be specified in title of this study. The authors should clarify this concern.
  2. The statistical methods used and described very well. Why type 3 Intra-class ICC for analysis?
  3. How does the authors to determine the sample size of this study? Please use power analysis to statement adequate sample size in this study.
  4. For tables, I suggested to add some tables of participants’ characters.
  5. We know the specific disease associated with relevant food intake. How is the progress related to diseases diagnosis among your proposed AI assessing food consumption? The authors should provide some information to the readers.
  6. Lastly, some references should be updated.

Author Response

  1. The study design should be specified in title of this study. The authors should clarify this concern.

We added the study design in the title : A cross-sectional reproductibility study of a standard camera sensor using artificial intelligence to assess food items: the FoodIntech project.

  1. The statistical methods used and described very well. Why type 3 Intra-class ICC for analysis?

We added a paragraph.

The estimation of the reliability of the AI results for each dish was assessed by the Type 3 Intra-class Correlation Coefficient (ICC) [25]. We used type 3 ICC issued from a two-way mixed model as recommended by Koo and Li 2016 [26]. Indeed, the reliability of a measurement refers to its reproducibility when it is repeated on the same subject. We wanted to have a complete overview with all possible camera angles, so positions where randomly selected among all the possible camera angles (infinite population). This corresponded to the random subject effect. Concerning repetitions, we were interested in assessing the 14 possible conditions, which reflect the finite, well-defined, possible measurement circumstances, and they were not randomly selected in an infinite population. This corresponded to the repetition factor, which was considered as a fixed effect in our model. All of these considerations were applied for each given food.

 These ICCs were determined using a random effect model from the irrNA R software package (version 4.1.0). It is important to note that the calculation of the ICC does not require the normality of the measurements and that the package allows the estimates to be made despite a varying number of conditions and images. The values of the ICC are between 0 and 1. The closer the ICC is to 1, the more similar the results are using AI under the same condition; however, the lower the ICC is (close to zero), the more the results of the AI diverge for the same condition. The central estimates for each ICC are accompanied by their 95% confidence interval, providing an estimate of the range in which the true value of the ICC could lie at the risk of a type I error of 5%. According to Shrout and Fleiss, an ICC greater than or equal to 0.8 supports excellent reliability; an ICC between 0.7 and 0.8 indicates good reliability. Values below 0.7 indicate moderate reliability at best [25; 27].

  1. How does the authors to determine the sample size of this study? Please use power analysis to statement adequate sample size in this study.

We added a paragraph.

The sample size of this study was determined using the PASS software. A sample size of 20 subjects (camera angles) with 14 repetitions (food conditions) per subject achieves 80% power. This can detect an intra-class correlation of 0.90 under the alternative hypothesis when the intraclass correlation under the null hypothesis is 0.80, using an F-test with a significance level of 0.05 [24].

  1. For tables, I suggested to add some tables of participants’ characters.

This study did not include participants. This is an experimental study in lab as indicated “The implemented method is based on the analysis of food tray pictures taken by a standard out-of-the box mobile phone (i.e. Android Samsung S8), before and after the simulated consumption of a standard meal tray extracted from the central kitchen of the Dijon University Hospital, and processed by laboratory research staff”

We added a paragraph in Materials and Methods 2.1. General procedure and change the title to clarify this point.

  1. We know the specific disease associated with relevant food intake. How is the progress related to diseases diagnosis among your proposed AI assessing food consumption? The authors should provide some information to the readers.

The aim of this study was to show the reproducibility of the system to detect food in a tray. In future work, the system will provide a measure of patient food consumption, which could be associated with age, gender and body weight. This information could make it easier for dietitians or physicians to adapt the menu. This is the aim of the last study finished in November, and data are under analysis for submission of one article in early 2022.

We added a chapter regarding future work:

Future work will aim to show the validity and the usability of the FoodIntech Artificial Intelligence system for the evaluation of food consumption in hospitalized patients. The system will provide a measure of patient food consumption, which could then be associated with the patient’s age, gender and body weight. Dietitians or physicians could use this information to adapt in-hospital menus to patient needs.

  1. Lastly, some references should be updated.

We have added more recent references in the introduction.

Reviewer 2 Report

The manuscript entitled ‘Assessing patient food consumption with a standard camera sensor using artificial intelligence: the FoodIntech project’ presents interesting issue, however some corrections are needed

  • The abstract should be a single paragraph and should follow the style of structured abstracts, but without headings
  • A sentence should never start with a number
  • Chapter ‘Discussion and conclusion’ should be divide into ‘. Discussion’ and ‘Conclusion’
  • Flow chart with some steps of procedures should be presented.
  • Moreover, the following steps of learning process should be presented – the results must be reproducible for other authors
  • There is no discussion, but rather repetition of the results (The repetition of results in the discussion section needs to be avoided). This section must be totally rewritten. Authors should relate the findings to those of similar studies and point the differences and similarities between the studies. Authors should add the appropriate references in this section.
  • Authors should present here and discuss the limitations of their study.
  • Precise conclusion should be formulated.
  • Figure 2 – Implementing computer vision technologies and a deep learning process – is difficult to follow
  • Authors should presents some results from this experiments – in the present form it rather some technical documents (however, detailed information is missing).

Author Response

  • The abstract should be a single paragraph and should follow the style of structured abstracts, but without headings

The headings were suppressed.

  • A sentence should never start with a number

We corrected the sentence in abstract.

  • Chapter ‘Discussion and conclusion’ should be divide into ‘. Discussion’ and ‘Conclusion’

We divided the chapter “Discussion and conclusion” in two chapters.

  • Flow chart with some steps of procedures should be presented.

We added a paragraph in Materials and Methods 2.1. General procedure to explain the procedures

  • Moreover, the following steps of learning process should be presented – the results must be reproducible for other authors

We added a paragraph in Materials and Methods 2.2 Algorithm used for Deep Learning, detailed

Figure 2 and added Appendix 2.

  • There is no discussion, but rather repetition of the results (The repetition of results in the discussion section needs to be avoided). This section must be totally rewritten. Authors should relate the findings to those of similar studies and point the differences and similarities between the studies. Authors should add the appropriate references in this section.

A discussion was added.

  • Authors should present here and discuss the limitations of their study.

The limitations of our study were put in the discussion.

  • Precise conclusion should be formulated.

Conclusions have been formulated.

  • Figure 2 – Implementing computer vision technologies and a deep learning process – is difficult to follow

Figure 2 has been simplified and better developed.

  • Authors should presents some results from this experiments – in the present form it rather some technical documents (however, detailed information is missing).

Detailed information and changes brought in this new version take into account of the comment of the reviewer.

Round 2

Reviewer 2 Report

I have no further comments

Author Response

We thank the reviewer